# The Potential Use of Cold-Pressed Pumpkin Seed Oil By-Products in a Low-Fat Salad Dressing: The Effect on Rheological, Microstructural, Recoverable Properties, and Emulsion and Oxidative Stability

**DOI:** 10.3390/foods10112759

**Published:** 2021-11-10

**Authors:** Zeynep Hazal Tekin-Cakmak, Ilker Atik, Salih Karasu

**Affiliations:** 1Faculty of Chemical and Metallurgical Engineering, Department of Food Engineering, Davutpasa Campus, Yildiz Technical University, Istanbul 34349, Turkey; zhazaltekin@gmail.com; 2Food Technology Program, Afyon Vocational School, Afyon Kocatepe University, Afyonkarahisar 03200, Turkey; ilkeratik@hotmail.com

**Keywords:** fat substitutes, emulsion stability, optimization, oxidative stability

## Abstract

The cold-pressed pumpkin seed oil by-product (POB) was evaluated for its application as a natural fat substitute and stabilizer in the reduced-fat salad dressings. For this aim, the samples were prepared by combining the xanthan gum (0.2–0.4 g/100 g), POB (1.0–5.0 g/100 g), egg yolk powder (3 g/100 g), and sunflower oil (10–30 g/100 g) in 17 different formulations. The optimization was carried out using response surface methodology (RSM) and full factorial central composite design (CCD). Results showed that all samples presented the shear-thinning (or pseudoplastic) flow behavior with 3.75–16.11 Pa·s^n^ and 0.18–0.30, K and n values, respectively. The flow behavior rheological data were fitted to a power-law model (R^2^ > 0.99). The samples with high POB and low oil content showed similar K and n values compared to high oil content samples. Additionally, the dynamic rheological properties and three interval thixotropic test (3-ITT) were determined. The G′ value was larger than G″ in all frequency ranges, indicating viscoelastic solid characteristics in all samples. The optimum formulation was determined as 0.384% XG, 10% oil, and 3.04% POB. The samples prepared with the optimum formulation (POBLF-SD) were compared to low-fat (LF-SD), and high-fat (HF-SD) control salad dressing samples based on the rheological properties, emulsion stability, oxidative stability, zeta potential, and particle size. The oxidation kinetic parameters namely, IP, E_a_, ΔS^++^, and ΔG^++^ showed that the oxidative stability of salad dressing samples could be improved by enriched by POB. The results of the present study demonstrated that POB could be considerably utilized as a natural fat substitute and stabilizer in salad dressing type emulsions.

## 1. Introduction

Salad dressing is a semi-solid oil/water (O/W) emulsion, mainly composed of vegetable oil (at least 30% by weight), stabilizers, emulsifiers, acidifying agents, salt, and flavorings. Oil is an important component of improved rheological properties, sensory characteristics, and other properties of the salad dressings [1]. Salad dressings are produced in three types according to the oil content, which are full fat (normal conventional fat level), low fat (minimum 25% fat), and lightweight (less than 1/3 fat or half fat) by the food industry [2]. When emulsion-based foods contain high fat, the amount of calories increases because fats have the highest number of calories per gram of the main food components. Due to health concerns, consumers want to reduce their calorie intake. However, it can be challenging for the food industry to produce low-fat products that satisfy customers’ expectations for fatty sensations while avoiding substantial changes in texture, rheological, and organoleptic qualities such as poor texture, taste, appearance, stability, and mouthfeel [3]. When producing low-fat salad dressing, oil substitutes with different functions should be used to replace the oil in the basic formula to obtain a product with the same quality characteristics as the original full-fat food products [4]. Food products that contained less fat may be undesirable for consumers because fat contributes to the texture, appearance, flavor, and the enhancing emulsion stability of emulsions [5,6]. Cold-pressed oil industry by-products have recently been utilized as fat substitutes [7,8,9].

In cold press oil extraction, there is less change in bioactive components in oils compared to refined oils since they are not subjected to high levels of heat and solvent treatment during their manufacturing. As a result, customer preference is focused on the cold-press extracted oils [10,11]. Since they are not subjected to chemical treatment and the food components in their structure are preserved, the use potential of the by-products generated after cold press oil production as a food source is high [12]. There is no trace of solvent in cold press oil by-products, which are rich in nutrients such as fat, protein and carbohydrates, dietary fiber. However, the yield is not as great as solvent extraction, and there are remarkable amounts of by-products after oil production in the cold press extracted oil industry. Recovery of these by-products and the utilization of them in food emulsions as a natural fat replacer, stabilizer, and emulsifier as well as an antioxidant source are important for food industry and environment. 

The pumpkin seeds may have 28–38 (*w/w*%) of oil consisting mainly of oleic, linoleic, and palmitic acids [13,14,15]. In addition to oil content, the considered amounts of proteins, crude fiber, minerals, phytosterols, tocopherols, and carotenoids (especially β-carotene and lutein) are found in seeds [16,17]. Pumpkin polysaccharides (PP) consist of galactose, glucose, arabinose, xylose, and glucuronic acid and water-insoluble substances with important biological function [18]. Wang, Cheng [19] isolated that a novel polysaccharide from pumpkin seeds (PSP-1) composed of mannose, glucose, and galactose with the main chain consisting of units of β →6)-β-D-Galp-(1→, →6)-α-D-Glcp-(1→, and →3,6)-β-D-Manp-(1→ with branching at O-3 and O-6 of →3,6)-β-D-Manp-(1→. Branch linkages were composed of α-D-Glcp-(1→ and →4)-α-D-Galp-(1 →). Dietary fiber serves to adjust the textural qualities of food, avoid syneresis, and stabilize high-fat food and emulsions by incorporating it into the product. [20].

Pumpkin seed oil is generally obtained by cold press processing so that it can be categorized as a group of very expensive and good-quality edible oils. Rabrenović, Dimić [21] studied the different bioactive compounds of cold-pressed pumpkin (*Cucurbita pepo* L.) seed oils obtained from different sources and they found that cold-pressed pumpkin seed oils were in various bioactive compounds such as monounsaturated fatty acids (37.1 ± 0.70–43.6 ± 0.69 g/100 g of total fatty acids), total tocopherols (38.03 ± 0.25–64.11 ± 0.07 mg/100 g of oil), sterols (718.1 ± 6.1–897.8 ± 6.8 mg/100 g of oil), and squalene (583.2 ± 23.6–747 ± 16 mg/100 g of oil). Pumpkin seed oil by-products may be also used in enhancing functional food products thanks to cold-pressed oil production conditions. According to the data obtained by Apostol, Berca [22], the partially defatted pumpkin seeds (12.28 g/100 g based on d.b.) were confirmed as a good source of bio-compounds, especially the total fiber (26.64 g/100 g based on db.) and could be utilized as potential prebiotic due to its fiber contents. Some studies have been conducted on the use of cold-pressed oil by-products in low-fat emulsions [8,9]. However, in this study, formulation optimization was performed for the first time in low-fat salad dressings prepared with the cold pressed pumpkin seed oil by-product (POB). In addition, in this study, the effect of POB on oxidation stability of low-fat salad dressing was firstly investigated. In this study, thermal loop test, zeta potential, and particle size distribution of low-fat salad dressing prepared with POB were also analyzed. In this study, a comprehensive characterization of low-fat salad dressing prepared with POB was performed.

In this study, the rheological properties (steady shear, dynamic rheological, and 3-ITT) of salad dressings stabilized by cold-pressed pumpkin seed oil by-products were studied in 17 different formulations. The main purpose of this study is to determine the optimum low-fat salad dressing sample containing POB that was carried out based on the shear steady rheological properties of the samples with different formulations. The low-fat (LF-SD) and high-fat control samples (HF-SD) were compared to samples produced with an optimum formulation (POBLF-SD) based on the steady shear, frequency sweep, and 3-ITT) rheological properties, oxidative stability, emulsion stability, light microscope images, and zeta potential and particle size. Thanks to these analyses, the possibilities of using POB in low-fat salad dressing (10% fat content) formulations will be determined as a natural stabilizer and fat replacer.

## 2. Materials and Methods

### 2.1. Material

For the preparation of the reduced-fat salad dressing samples, POB was obtained from ONEVA Food Co. (Istanbul, Turkey). POB is the press cake released after cold press extraction of pumpkin seed (*Cucurbita pepo* L.). The shells of the pumpkin seeds were removed before pressing. The press temperature did not exceed 50 °C during the cold-press process. POB contained 94.30% dry matter, 9.00% oil, 44.02% protein, 37.81% carbohydrate, and 3.47% ash. POB was ground using a laboratory mill (PX-MFC 90 D, Kinematica, Malters, and Switzerland) and sieved through mesh No.140. Sunflower oil was purchased from a local market. Xanthan gum (XG) and egg yolk powder (EYP) were obtained from Sigma-Aldrich (Sigma Chemical Co., St. Louis, MO, USA). All emulsions were prepared by using double distilled water. The chemicals and standards were analytical grade and purchased either from Merck (Darmstadt, Germany) and Sigma-Aldrich (St. Louis, ABD).

### 2.2. Methods

#### 2.2.1. Salad Dressing Preparation

POB-enriched salad dressings were prepared by using the following ingredients which are XG, POB, EYP, and oil expressed as g/100 g. All formulations of the salad dressing samples were prepared according to the experimental design shown in Table 1. The method of the preparation was as follows: POB were slowly added to water to form POB– water dispersion and then the dispersion heated to 80 °C for 20 min to obtain a complete hydration of POB in water. After cooling the solution to 25 °C, XG and EYP were dissolved in the dispersion. The stirring of the mix was continued at 1000 rpm in a magnetic stirrer for 6 h to complete the hydration of the XG. Finally, an Ultra-Turrax digital homogenizer (Daihan, HG-15D, Gangwondo, Korea) was used for the preparation of O/W emulsions at 10,000 rpm for 3 min [23]. Finally, the salad dressing samples were obtained and pasteurized at 65 °C for 10 min and poured into brown bottles and cooled to room temperature. In this study, all material (beakers, brown bottles, and probes) was used after the sterilization 121 °C for 15 min. High fat (HF-SD) and low fat control salad dressing samples (LF-SD) were prepared. HF-SD and LF-SD samples were prepared with 30% and 10% oil, respectively. XG and EYP were added as 0.35% and 3% in both control samples, respectively. The pH of the HF-SD and LF-SD samples were 3.59 and 3.75, respectively. A grape vinegar containing 4% acetic acid at the rate of 7.5% of the aqueous phase was used for acidification of the products.

#### 2.2.2. Experimental Design

The effect of different formulations of XG (g/100 g), oil (g/100 g), and POB (g/100 g) on the preparation of the reduced-fat salad dressing samples were determined according to response surface methodology and full factorial central composite design. As presented in Table 1, 17 different experimental points were obtained by using Design Expert Software (Version 7; Stat-Easy Co., Minneapolis, MN, USA) to determine the optimum formulation. The design comprised three of the factorial points, for the estimation of the error. Model applicability was evaluated based on the R^2^, R^2^-adj, lack of fit, F, and P-values obtained from ANOVA. Based on the highest desirability value, the optimization was carried out. The formulation with the lowest fat content with a desirability value of 1 was determined as the optimum formulation. Three central points were used. Analysis of all points was conducted in triplicate. Additionally, the results were reported as mean values and standard deviations.

#### 2.2.3. Rheological Properties

The salad dressings’ rheological characteristics (flow behavior, dynamic rheological, and 3-ITT) were determined at 25 °C using a stress or strain and temperature-controlled rheometer (Anton Paar, MCR 302, Graz, Austria) equipped with a Peltier system. PP50 probe (diameter of 50 mm) was utilized as measurement geometry and a gap height of 0.5 mm was preferred for measurements. All rheological studies were performed in triplicate at 25 °C. Before starting the test, all samples were allowed to equilibrate at 25 °C for 1 min.

##### Steady-Shear Rheological Properties

The flow behavior in the rheological characteristics of salad dressings was investigated at shear rates ranging from 0.01 to 100 s^−1^. The shear stress and apparent viscosity parameters associated with the shear rate were measured. The experimental flow curves were fitted to the power-law model expressed in Equation (1), and the relevant parameters of steady shear rheological characteristics, K and n value, were determined.
(1)$$\tau =K\gamma^{n}$$
where τ was shear stress (Pa), K was consistency index (Pa·s^n^), γ was shear rate (1/s), and n was the flow behavior index. 

##### Dynamic Rheological Properties

For the dynamic rheological analysis, a frequency sweep test was conducted. Prior to the oscillatory measurements, an amplitude (stress) sweep test with a strain value of 0.1% was performed to identify the linear viscoelastic area (LVR). In LVR, the frequency sweep test was performed in the 0.1–64 (ω) angular velocity range. In response to the rotational velocity, the storage (G′) and loss modulus (G″) values were calculated. The power-law model and nonlinear regression were used to compute the dynamic rheological parameters (G′ and G″) [24];
(2)$$G^{\prime }=K^{\prime }{\left(\omega \right)}^{n^{\prime }}$$
(3)$$G^{\prime \prime }=K^{\prime \prime }{\left(\omega \right)}^{n^{\prime \prime }}$$
where G′ (Pa) is the storage modulus, G″ (Pa) is the loss modulus, ω is the angular velocity value (1/s), and K′ (Pa·s^n^) and K″ (Pa·s^n^) values indicate the consistency index values, and n′, n″ values represent the flow behavior index values. 

##### The Three Interval Thixotropy Test (3-ITT)

3-ITT was applied to the salad dressing samples to obtain information about the degree of recovery after deformation. The 3-ITT rheological properties of the samples were respectively determined to be at the constant shear rate of 0.5 s^−1^ and a variable shear rate of 150 s^−1^. When the values were selected, the linear viscoelastic region was considered, and the linear viscoelastic region of the samples ends at 50 s^−1^. The samples were subjected to 100 s at a very low shear rate (0.5 s^−1^) in the first time interval and subjected to high shear rate (150 s^−1^) in the second interval for 40 s. In the third time interval, the samples were exposed to the shear rate applied in the first time interval and their recovery was tested. For this purpose, the change in G′ value was recorded. The solid like behavior of the samples in the third time interval was modeled using a second-order structural kinetic model:
(4)$${\left[\frac{G^{\prime }-G_{e}}{G_{0}-G_{e}}\right]}^{1-n}=(n-1)kt-1$$
where k is the thixotropic rate constant, G_0_ is an initial storage modulus (Pa) in the third time interval, and G_e_ is the equilibrium storage modulus (product completely recovers itself) [25].

#### 2.2.4. Analysis of Optimum and Control Samples

##### Rheological Analysis

The rheological properties (flow behavior, dynamic rheological, and 3-ITT) of the salad dressings (control samples and the optimum sample contained 3% POB) were determined at 25 °C using stress or strain and temperature-controlled rheometer (Anton Paar, MCR 302, Austria) equipped with a Peltier system.

##### Emulsion Stability by Thermal Loop Test

Emulsion stability of salad dressing samples (control samples and the optimum sample contained 3% POB) was determined by the thermal loop test previously described by Tekin, Avci [26]. The samples were subjected to ten thermal cycles from 23 to 45 °C in a high-temperature stability test. An angular frequency (ω) and strain values were 10 Hz and 0.5%, respectively. The heating and cooling rates were set at 11 °C/min. The maximum points of all cycles were recorded by using the rheometer software (Rheoplus for MCR 301) using the internal loop. 

##### Oxidative Stability by OXITEST

The Oxitest Device (Velp Scientifica, Usmate, MB, Italy) was used to test the oxidative stability of the salad dressing samples (control samples and the optimal sample containing 3% POB) during the storage period [27]. In total, 10 g of salad dressing sample was put into the sample cells to ensure uniform distribution. The temperature of the device was adjusted to 80, 90, 100, and 110 °C, and the oxygen pressure was set to 6 bar. The induction period (IP) value obtained by the OXITEST equipment was used to evaluate the oxidative stability values of the samples. In addition, depending on the period, data on pressure change was received from OXITEST. The impact of temperature on the oxidation rate constant (k), which is the inverse of IP, was determined using the Arrhenius equation (Equation (5)):(5)$$k=A_{0}\times \exp\left(\frac{-E_{a}}{RT}\right),$$
where E_a_ represents the activation energy (kJ/mol), R shows the ideal gas constant (8314 J/molK), and T is the temperature (K). Activation enthalpy (ΔH^++^) and entropy (ΔS^++^) values were calculated by using Equation (6) derived from the activated complex theory:(6)$$k=\frac{k_{B}t}{h}\exp\left(-\frac{\Delta H^{++}}{RT}+\frac{\Delta S^{++}}{R}\right)$$

In this equation, kB represents the Boltzmann constant (1.3806488 × 10^−23^ J/K), h represents the plank constant (6.6261 × 10^−34^ J/s), T is the absolute temperature (K), R is the ideal gas constant (8314 J/mol.K), ΔH^++^ and ΔS^++^ are the enthalpy (kJ/mol) and entropy change (J/mol/K), respectively. Equation (7) was used to determine the required parameters using the nonlinear regression model. Statistica software (StatSoft, Tulsa, OK, USA) was used for nonlinear regression. The free energy of activation (ΔG^++^)
(7)$$\Delta G^{++}=\Delta H^{++}-T\Delta S^{++}$$

##### Zeta Potential and Particle Size

The size of the fat globules and zeta potential value of the samples was determined by the particle size measuring device (Nanosizer, Malvern Instruments, Worcester, UK). The samples were diluted 500-fold with ultrapure water and homogenized by sonication in an ultrasonic water bath for 1 min. The dynamic light scattering technique was used to determine the particle size of the samples [26].

##### Optical Microscopy

The optical observation of salad dressing samples (control samples and the optimum sample contained 3% POB) was conducted by a light microscope equipped with a digital camera (Cannon, Japan). Images were analyzed using a specific software program. About 0.01 mL emulsion samples before and after US treatment were placed on a glass slide and covered with a coverslip. After 30 min of equilibration, samples were analyzed at 26 °C at 100 μ magnification.

#### 2.2.5. Statistical Analysis

Salad dressing samples were produced in triplicates and each replication was subjected to three parallel measurements. The mean and standard deviation of the data are shown in tables. The Statistica software program (Stat Soft Inc., Tulsa, UK) was used for statistical applications. The mean values of the parameters were compared using Duncan’s multiple comparison tests (*p* < 0.05). Power-law and second-order structural kinetic model parameters were generated using nonlinear regression analysis as a consequence of the applied rheological study. Additionally, oxidation kinetic parameters were calculated by nonlinear regression. Nonlinear regression analysis was performed using the Statistica software package (Stat Soft Inc., Tulsa, UK).

## 3. Results and Discussion

### 3.1. Rheological Properties

#### 3.1.1. Steady-Shear Rheological Properties

Figure 1 shows the shear rate versus shear stress curves exhibiting non-Newtonian flow behavior for the salad dressing samples containing 17 different levels of XG, POB, and oil. As seen, the slope of the figure decreased with increasing shear rate, showing that all samples showed shear-thinning (or pseudoplastic) behavior. Lai and Lin [28] also reported that the low-fat salad dressing model emulsions showed the shear thinning flow behavior, which indicates a decrease in the viscosity values of salad dressing samples due to the increased shear rate. The decrease in viscosity values of salad dressing type food emulsions can be explained by the result of a dramatic structural breakdown in intermolecular bonds as previously reported for other food emulsions [29,30,31,32,33]. 

In Figure 1, when the curves of POB-10 (30% oil, 0.4% XG, 5% POB) and POB-2 (30% oil, 0.4% XG, 1% POB) were examined, it was observed that an increase in the amount of POB from 1% to 5% at the same oil and XG values allowed a significant increase in the viscosity value. The curves of the POB-3 sample (20%, 0.4% XG, 3% POB) and the POB-15 sample (10%, 0.4% XG, 5% POB) were almost overlapping, indicating that these samples showed the similar steady shear rheological properties. The reason why the curve is similar with decreasing the oil amount from 20% to 10% can be explained by increasing the amount of POB from 3% to 5%. In salad dressing type emulsions, the reduction of oil content could change their rheological and textural behavior and has also profoundly influenced their stability during storage. Therefore, the fact that a low-fat salad dressing with POB (although added at low concentrations) and a high-fat salad dressing show similar rheological properties is an important development for the food industry

Power-law model parameters (K and n values) and determination coefficient (R^2^) calculated for 17 different points created with the trial design are shown in Table 1. R^2^ values of the power-law model were higher than 0.97, showing that the power-law model was suitable for determining the steady shear rheological properties of salad dressing samples. Consistency, an important quality parameter for emulsion-type foods such as purees, pastes, sauces, and salad dressings, indicates a strong interaction between the molecules in the sample structure and stability and shows the thickness of the emulsions [34]. The consistency index (K) values were determined by Equation (1) and found as between 3.75 and 16.11 Pa·s^n^ (Table 1). As can be observed from the table, K increased with increasing XG (g/100 g) and/or with increasing POB (g/100 g). The K value increased with the increase of POB (g/100 g) because the high branched polysaccharide of POB provided the high water-holding ability [19]. Optimization was examined based on the desired K value of 8.00 Pa·s^n^ in a salad dressing with 30% oil (full-fat salad dressing). Considering this value, when the amount of POB was increased from 1% to 5% in POB-5 and POB-15 samples containing 10% oil, the K value increased from 6.41 to 8.95 Pa·s^n^. With this situation, it was concluded that with the addition of more than 1% POB, low-fat salad dressing could have a consistency index value similar to full-fat salad dressing. These results showed the significant increase in consistency. The flow behavior index (n) for food emulsions can be associated with the mean particle size (oil droplets), the particle size distribution, and the colloidal nature of the continuous phase [28]. The n values for all salad dressing samples obtained by fitting the data to the power-law model (Equation (1)) were less than 1 (indicative of pseudo-plastic (shear-thinning) and non-Newtonian behavior) and ranged from 0.17 to 0.81 (Table 1). The non-Newton flow characteristics (Herschel Bulkley) and the low flow behavior index value (n) (the more viscosity dependence to the shear rate) are the desired flow behavior properties of salad dressings when salad dressings have the desired degree of consistency and yield stress (yield stress) [6]. Additionally, a decrease in the value of n was observed as the consistency coefficient increases as expected. The n value of the samples with a low consistency coefficient was found to be greater. The negative relationship between the K and n values can be explained by the increase in the shear thinning or pseudoplastic flow behavior and the formation of a tight structure. 

#### 3.1.2. Dynamic Rheological Properties

The dynamic oscillatory shear test was used to characterize the viscoelastic properties of the salad dressing samples and determine the storage (G′) and the loss (G″) moduli shown in Figure 2. It was observed that G′ values were higher than G″ values in all salad dressing samples, meaning that all the salad dressings were found to be more elastic than viscous. In the literature, the mayonnaise is a viscoelastic system like salad dressing, and the G′ values of samples are higher than G″ values [35,36]. The G′ values of the samples show solid character [37]. Therefore, when the oil content increases in the salad dressing samples, the G′ values of samples increase. As seen in Figure 2, all salad dressing samples contained higher XG, POB, and oil content showed higher G′ value. It can be explained that POB improved desired solid-like nature of the salad dressing samples. The dietary fiber content of POB helps to modify the textural properties of the salad dressing samples, avoid syneresis, and stabilize high fat [20].

The dynamic rheological analysis data were obtained by using the power-law model, and K′, K″, n′, and n″ values were obtained by using nonlinear regression (Table 2). As can be seen from Table 2, R^2^ values were found to be high (R^2^ > 0.93) and the high value of R^2^ shows that the model can successfully explain the dynamic rheological behaviors of the samples. As can be seen from the table, K′ and K″ values of the samples were estimated as 0.64–93.40, and 0.94–28.88 Pa·s^n^, respectively, and these values changed according to the formulation used.

#### 3.1.3. The 3-ITT Rheological Properties 

Figure 3 shows the 3-ITT rheological properties of the salad dressing samples. The thixotropic behavior is important for O/W emulsions especially in salad dressings and other emulsions containing low and medium oil due to the deformations during high-speed mixing and homogenization, as well as during consumption, such as when the packed food is shaken or squeezed. As can be seen in Figure 3, all samples showed thixotropic behavior in the third interval. After high shear deformation, samples lost their viscoelastic properties and then regained them in a third interval. These findings suggested that all salad dressing samples may maintain their viscoelastic character throughout food processing involving a large amount of abrupt deformation, such as homogenization or pumping, as well as consumption under shaking and squeezing. This is the ideal flow behavior for salad dressing emulsions [25]. Akcicek and Karasu [7] studied the 3-ITT rheological properties of the salad dressing utilized by chia seed oil by-product. Tekin and Karasu [8] reported that flaxseed oil by-products improved the thixotropic behavior of the salad dressing samples after sudden deformation. These researchers reported recoverable properties in the third interval that are similar to our study. 

Table 3 shows the second-order structural kinetic model parameters (G_0_, G_e_, k) obtained by Equation (4) and k × 1000 value and the ratio of equilibrium G_e_ and initial G_0_ (G_e_/G_0_). G_0_, G_e_, G_e_/G_0_, k, k × 1000, and R^2^ values were 5.61–77.24, 5.0–81.09, 0.74–1.33, 0.01–0.13, 9.33–128.55, and 0.95–1.00, respectively. POB-10 sample (30% oil, 0.4% XG, 5% POB) showed the highest G_0_, G_e_, G_e_/G_0_, k × 1000 values, indicating that POB-10 showed the highest thixotropic behavior and viscoelastic solid character. Therefore, the sample showed the highest thixotropic behavior. The amount of fat in an O/W emulsion has a significant impact on its rheological characteristics. Therefore, the decrease in the fat content of the salad dressing samples causes weak rheological properties, especially the recoverable character, as the fat content of the salad dressing samples decreases. As can be seen in Table 3, POB-1 and POB-17 have close G_e_/G_0_ (1.23 and 1.21, respectively) values, indicating that although the oil is reduced by 10%, increasing the amount of POB from 3% to 5% allows it to exhibit similar thixotropic properties. Therefore, POB can be utilized to enhance the rheological properties and thixotropic properties of low-fat salad dressing samples. Increased intermolecular contacts caused by the development of small hydrocolloid aggregates can explain the greater recoverable feature when POB and XG concentrations rise. By increasing the high shear force delivered in the second time interval, these interactions and aggregation may be disrupted. This interaction might develop during the third time interval when the applied shear forces are minimal. For the 3-ITT test, forming this internal structure at lower shear stress indicates strong thixotropic behavior and recoverability [38].

#### 3.1.4. The Effect of Model Parameters on K Value and Determination Optimum Formulation

The effect of XG, POB, and oil in the formulation on K value is shown in Figure 2. As demonstrated in Figure 4, an increase in XG, oil, and POB resulted in an increase in the K value of the samples. This effect can be explained by the chemical structure of POB, which comprises high protein and polysaccharides with strong water-holding and surface-active ability. In addition to its stabilizer properties due to polysaccharides, POB proteins may be absorbed in the interface and have surface-active properties. The K value of the salad dressing samples increased as a result of these properties of protein and polysaccharides of POB. Furthermore, increasing the amount of gum resulted in a significant increase in the K value, especially in formulations including a specific amount of xanthan gum. The synergistic effect of the components used in salad dressing formulation can explain the increase in K value in both increases in XG, foil, and POB.

The quadratic model was used to determine the effect of formulation parameters on the K value of the salad dressing samples. The model’s R^2^, Adjusted R^2^, and predicted R^2^ values were determined to be 0.9677, 0.9262, and 0.8460, respectively (Table 4). The gap between Adjusted R^2^ and a predicted R^2^ was less than 0.2, and the lack of fit was insignificant. The adequation precision was 18.77. These findings suggested that the quadratic model might be used to accurately characterize the effect of formulation on sample K values. The model’s *p*-value was less than 0.05, indicating that the model terms had a substantial impact on the K value. A, B, C, AC, B^2^, and C^2^ are important model terms in this model. The linear influence of all independent variables had a significant effect. The interaction and model terms A and C, and the quadratic effect of B and C were also be found as significant. These results showed that POB significantly affected the K value of the salad dressing. The interaction of the POB and oil also be found significant. The optimum formulation was determined as 0.384% XG, 10% oil, and 3.04% POB. 

#### 3.1.5. Analysis of HF-SD, LF-SD, and POBLF-SD

##### The Rheological Properties of HF-SD, LF-SD, and POBLF-SD

The formulation optimization was based on the K value of the HF-SD (30% of oil, 0.35% XG, and 3% EYP). The K value of HF-SD, which was chosen as the target value to determine the optimum salad dressing formulation, was determined as 8.02 Pa·s^n^. In this study, the aim was to obtain the optimum low-fat salad dressing (POBLF-SD) by using POB with properties similar to high-fat salad dressing. For this purpose, the minimum oil content (10%) with desirability value of 1.00 (showing K value of full fat control samples) were chosen as criteria to determine the optimum formulation. The formulation of POBLF-SD was 10% of oil, 0.36% of XG, and 3.04% of POB. We produced optimum (POBLF-SD) and control samples (HF-SD and LF-SD) and compared them to validate the experimental data and RSM results. Table 5 shows the steady-shear, dynamic, and thixotropic properties of HF-SD, LF-SD, and POBLF-SD. K value of POBLF-SD 8.21 Pa·s^n^ was determined as showing that the correlation between the experimental and predicted data was high and the response model successfully described the optimization process. 

The flow behavior of salad dressing samples can be interpreted through shear stress vs. shear rate curves (Figure 5). As can be seen, HF-SD, LF-SD, and POBLF-SD samples showed shear-thinning (pseudoplastic) rheological behavior. The shear-thinning behavior of salad dressing samples, which follows a similar behavior, was fitted to the power-law model [8,28,39]. The power-law model parameters (K and n values) and the determination coefficient (R^2^) were between 3.78–8.21 Pa·s^n^, 0.19–0.23, and 0.99–1.00, respectively (Table 5). These parameters of HF-SD and POBLF-SD samples were similar so that POB can be used as a fat substitute for salad dressing samples. This result showed that POBLF-SD and HF-SD (30% oil content) showed similar rheological behavior thanks to the addition of 3.04% POB, although POBLF-SD contained 20% less oil. Therefore, the polysaccharides in POB act as emulsion stabilizers by forming an extended network in the continuous phase, increasing its viscosity, and the proteins in the POB behave as emulsifiers by adsorbing at the oil–water interface. Therefore, POB can be preferred as a stabilizer and an emulsifier for low-fat salad dressing samples. Tekin-Cakmak, Karasu [9] suggested that cold-press byproducts (black cumin seed (BOB), coconut (COB), flaxseed (FOB), and pumpkin seed (POB)) may be utilized in salad dressings as natural fat replacements and functional additives. In this study [9], all by-products (FOB, POB, COB, and BOB) increased the pseudoplastic and viscoelastic solid character of low-fat salad dressing samples and may be utilized as a natural fat substitute in low-fat salad dressing at a specific ratio (FOB and POB 3%; BOB and COB 1%).

The dynamic rheological behavior of salad dressing samples prepared according to optimum, high-fat, and low-fat control formulations is shown in Figure 4. In salad dressing samples, increasing G′ and G″ values of samples with increasing frequency is evidence of gel-like behavior [8]. As seen in Figure 4, the G′ value of all samples was higher than the G″ value, indicating that the solid character of all salad dressing samples is more dominant than the liquid character. Additionally, the G′ values of HF-SD (30% oil) and POBLF-SD (10% oil) samples were almost at the same level, explaining that 20% oil can be compensated with 3.04% POB (Figure 4). As can be seen, the LF-SD has the lowest G′ and the lowest G″ values. The G′ and G″ values of the LF-SD (10% oil, 0.35% XG) were lower than the G′ and G″ values of the POBLF-SD (10% oil, 0.36% XG, 3.04% POB). As it can be also seen in Table 5, POBLF-SD has the elastic modulus value as much as HF-SD with a high-fat content (30%). Synergetic interactions between POB and other salad dressing ingredients can lead to improved food quality and expanded food applications due to enhanced functional properties. The dynamic rheological parameters (K′, K″, n′, and n″ values) were also calculated by using the power-law model (Table 5). The *R*^2^ values of the model were found in the range of 0.82–1.00. As can be seen in Table 5, the K′ and K″ values of the samples were in the range of 5.35–15.78 and 1.20–6.16, respectively; the values of n′ and n″ were found in the range of 0.14–0.36 and 0.09–0.37, respectively. The K′ values were higher than the K″ values for all samples. Accordingly, all of the salad dressing samples showed a viscoelastic solid character. When POB was added to the salad dressing sample, K′ and K″ values of POBLF-SD were also higher than the HF-SD sample’s values. 

Figure 5 exhibits that all salad dressing samples show the thixotropic behavior in the third interval, meaning that all samples could recover their viscoelastic character after high sudden deformation during food processing, such as homogenization or pumping. This thixotropic behavior is desirable for salad dressing samples. Akcicek and Karasu [20] investigated the thixotropic behavior of salad dressing samples stabilized by chia seed oil by-products and found that recoverable characteristics in the third interval are similar to our findings. Therefore, POB can be also used as a stabilizer for salad dressing samples. Table 5 also shows that 3-ITT parameters (G_0_, G_e_, k, G_0_/G_e_) were obtained with the second-order structural kinetic model. G_0_, G_e_, k, G_e_/G_0_, k × 1000, and R^2^ values were 6.87–17.91, 8.00–22.85, 0.04–0.06, 1.16–1.23, 43.32–56.64, and 0.98–0.99, respectively. POBLF-SD showed the highest G_0_, G_e_, k, G_e_/G_0_, k × 1000 values, indicating that the POBLF-SD sample showed the highest thixotropic behavior. The amount of oil in an O/W emulsion has a significant impact on its rheological characteristics. This result showed that the addition of POB may prevent the changes in the rheological properties caused by the reduction of oil. The higher recoverable behavior obtained with POB addition can be explained by more intermolecular interactions by the formation of small aggregates of hydrocolloid. These rheological properties indicate that POB can be utilized to enhance the rheological and thixotropic properties of low-fat salad dressing samples.

##### Emulsion Stability, Zeta Potential and Particle Size, and Microstructure of HF-SD, LF-SD, and POBLF-SD

The emulsion stability of the salad dressing samples was determined by the thermal loop test [26]. The thermal loop test could be used as a fast method to measure emulsion stability by subjecting to different numbers of thermal cycles. By the thermal loop test, the temperature fluctuations were simulated during the processing, production, storage, and transportation stages. The structural or morphological changes due to the applied thermal stress were determined by the change of modules (G*, G′) from cycle to cycle. Figure 6 shows the change in G* value after applied thermal stress. As can be seen, after 10 applied thermal cycles, a slight increase in the G* value of all samples was observed for the samples POBLF-SD and HF-SD. A dramatic increase in the 10th thermal cycle was observed in the G* value of LF-SD, indicating that LF-SD had lower emulsion stability than other samples. The insignificant change in POBLF-SD and HF-SD indicates that these samples were resistant to thermal stress and have high emulsion stability.

The zeta potential values, oil particle size (d_32_), and PDI value of the samples were found as (−43.15)–(−39.68) mV, 3125.67–51.96.00 μm, and 0.27–0.90, respectively (Table 5). As can be seen, the sample containing POB exhibited lower particle size and PDI value. These results are similar to salad dressing studies in the literature [7,8]. The adsorption of negatively charged polysaccharides onto the negatively charged proteins’ surfaces via various interactions such as H-bonding and hydrophobic contact resulted in higher negative zeta potential. Salad dressing samples with the greatest zeta potential were found to have a value of (−43.15)–(−39.68) mV. Salad dressing samples with high zeta potential values may have been anticipated to have long-term stability [8]. These results are consistent with the thermal loop test. A decrease was observed in the zeta potential values of the samples as the water ratio increased. However, all samples have a sufficient zeta potential value. 

Figure 7 shows the oil particle distribution. The POBLF-SD sample has homogeneous droplet distribution and low oil droplet size; however, the HF-SD sample has droplet density and size increased with the non-homogenous distribution. These results indicated that the use of POB could have a positive effect on the fat droplet size and distribution and emulsion stability in low-fat salad dressing samples.

##### Oxidative Stability

The IP values of the salad dressing samples are shown in Table 6. The IP values of HF-SD, LF-SD, and POBLF-SD were found as 1.03–12.57, 0.45–10.34, and 1.29–16.25 h respectively. The HF-SD sample showed a higher IP value than the LF-SD sample, indicating that the decrease in the oil ratio in salad dressing led to a decrease in the IP value. POBLF-SD showed a higher IP value than HF-SD and LF-SD, indicating that POBLF-SD showed the highest oxidative stability.

Arrhenius and Eyring’s equations modeled the effect of temperature on the oxidation kinetics of the samples by using nonlinear regression. Table 6 presented the Arrhenius and Activation of complex parameters. As shown, sample type significantly affected E_a_, ΔH^++^, ΔS^++^, and ΔG values (*p* < 0.05). The E_a_ value shows the minimum energy value required to initiate oxidation. E_a_ values were 87.63, 76.69, and 88.83 kJ/mol for HF-SD, LF-SD, and POBLF-SD, respectively, indicating that the addition of POB significantly increased the E_a_ values of the samples compared to LF-SD. AKSOY, TEKIN-CAKMAK [27] reported that the addition of some microencapsulated phenolic extracts significantly increased the E_a_ value of the salad dressing samples. Farhoosh [40] reported that the antioxidant type differently affected E_a_ of the oil oxidation. The activation complex parameters, namely, ΔH^++^, ΔS^++,^ and ΔG^++^ were used to evaluate the effect of temperature on the oxidation kinetic of salad dressing samples. ΔH^++^ values of the HF-SD, LF-SD, and POBLF-SD were calculated as 87.63, 76.69, and 88.83, respectively while the ΔS^++^ values were found as 6.61, 22.35, and −28.84, respectively. Similar ΔH^++^ and ΔS^++^ values were reported for salad dressing and oil oxidation [41,42,43]. As can be seen, the low fat and high-fat control samples showed positive ΔS^++^ values while samples formulated with POB showed negative ΔS^++^ values. The positive value of ΔH^++^ indicates the endothermic behavior of activated complex formation. The strong negative value of ΔS^++^ indicated that the activated complexes are more ordered than the reactants molecules, and fewer numbers of species in the activated complex state [42]. The negative value of ΔS^++^ obtained from POBLF-SD indicates a slower oxidation rate. The decrease in ΔS^++^ with the addition of POB can be explained by the reduction of free radical concentration by their hydrogen donation of POB antioxidants and loss of rotational freedom in the transiently activated complex [41]. 

ΔG^++^ value was calculated to obtain quantitative information about the oxidation rate. The higher value of ΔG^++^ is the indication of a slower oxidation rate and strong oxidation stability. ΔG^++^ values of HF-SD, LF-SD, and POBLF-SD were 86.85–87.05, 85.97–86.64, and 88.04–88.91 kJ/mol, respectively, indicating that POBLF-SD showed the highest ΔG^++^ value. The results of the IP value, the Arrhenius, and the Activation of complex parameters were compatible with each other. These results indicated that the oxidative stability of low-fat salad dressing samples could be improved with the addition of POB as well as the improvement of rheological properties. The increase in oxidative stability by the addition of POB can be explained by the free radical scavenging activity of the phenolic antioxidant of POB, which is localized to the oil–water interface.

## 4. Conclusions

The main aim of this study was to investigate the potential use of POB as a fat substitute that can be used to obtain low-fat salad dressing with similar rheological, microstructural, recoverable properties, and emulsion and oxidative stability to high-fat salad dressing. The addition of POB considerably contributed to these properties of the salad dressing due to being rich in protein and carbohydrates. All samples with 17 different formulations exhibited shear-thinning and viscoelastic solid character. Steady-state and viscoelastic flow behavior of the salad dressing formulated with low oil and 3% POB content (10% oil, 0.3% XG, 3% POB) showed similar characteristics with high oil content salad dressing. These results indicated that salad dressing samples formulated with low oil and 3% POB content showed desirable recoverable character and could preserve their viscoelastic nature during the process with a high sudden deformation. The low-oil sample containing POB (POBLF-SD), produced as a result of the optimization, showed similar rheological properties to the high-oil sample (HF-SD). The free radical scavenging activity of POB’s phenolic antioxidant, which is localized to the oil–water interface, might explain the improvement in oxidative stability caused by its addition. This study suggested that POB could be successfully used as a natural fat replacer, stabilizer, and oxidative agent in a variety of food applications.

## Figures and Tables

**Figure 1 foods-10-02759-f001:**
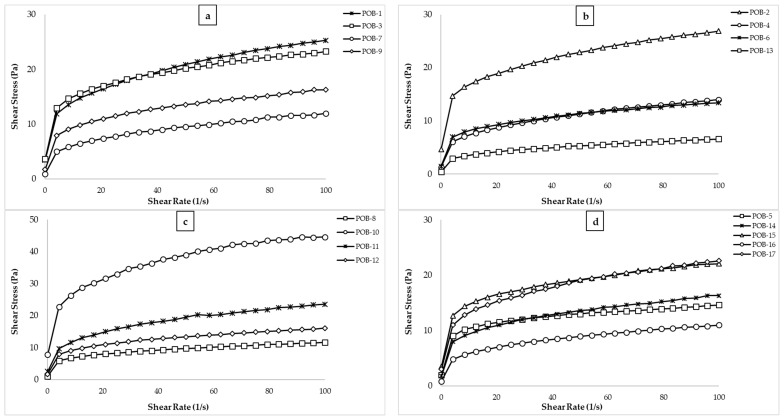
Steady shear rheological properties of the salad dressing samples contained with POB (POB: cold-pressed pumpkin seed oil by-product). (**a**): POB-1, 3, 7, & 9; (**b**): POB-2, 4, 6, & 13; (**c**): POB-8, 10, 11, & 12; (**d**): POB-5, 14, 16, & 17.

**Figure 2 foods-10-02759-f002:**
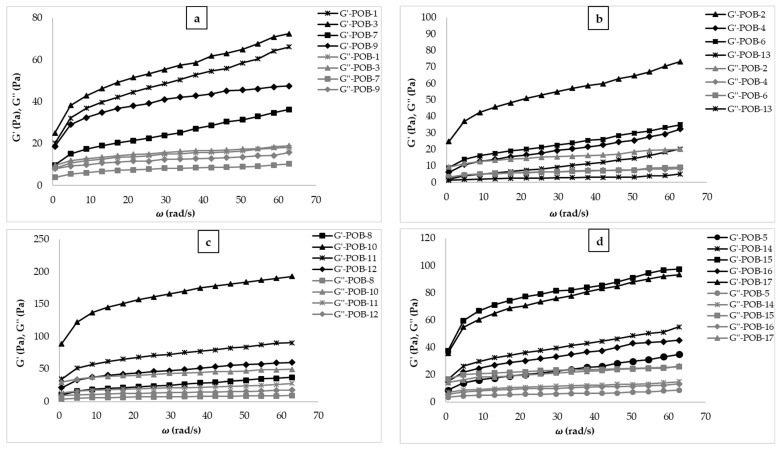
G׳ and G″ values versus angular velocity values of salad dressing samples (POB: cold-pressed pumpkin seed oil by-product, G′ (Pa): storage modulus, G″ (Pa): loss modulus). (**a**): G′ and G″ of POB-1, 3, 7, & 9; (**b**): G′ and G″ of POB-2, 4, 6, & 13; (**c**): G′ and G″ of POB-8, 10, 11, & 12; (**d**): G′ and G″ of POB-5, 14, 16, & 17.

**Figure 3 foods-10-02759-f003:**
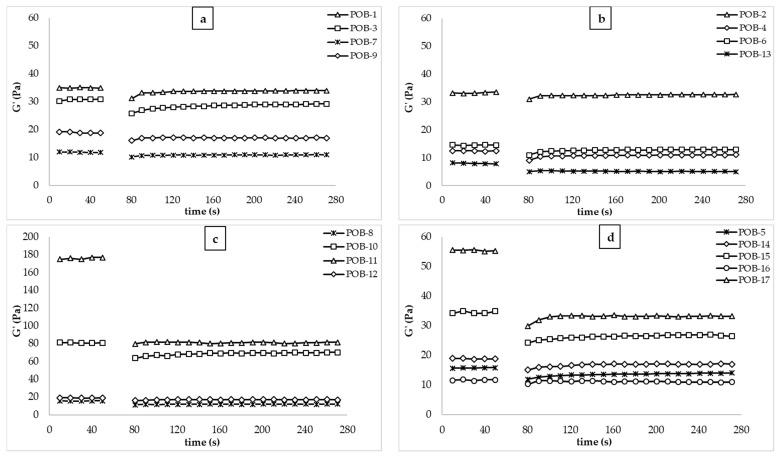
3-ITT rheological properties of salad dressing samples (POB: cold-pressed pumpkin seed oil by-product, G′ (Pa): storage modulus)). (**a**): G′ of POB-1, 3, 7, & 9; (**b**): G′ of POB-2, 4, 6, & 13; (**c**): G′ of POB-8, 10, 11, & 12; (**d**): G′ of POB-5, 14, 16, & 17.

**Figure 4 foods-10-02759-f004:**
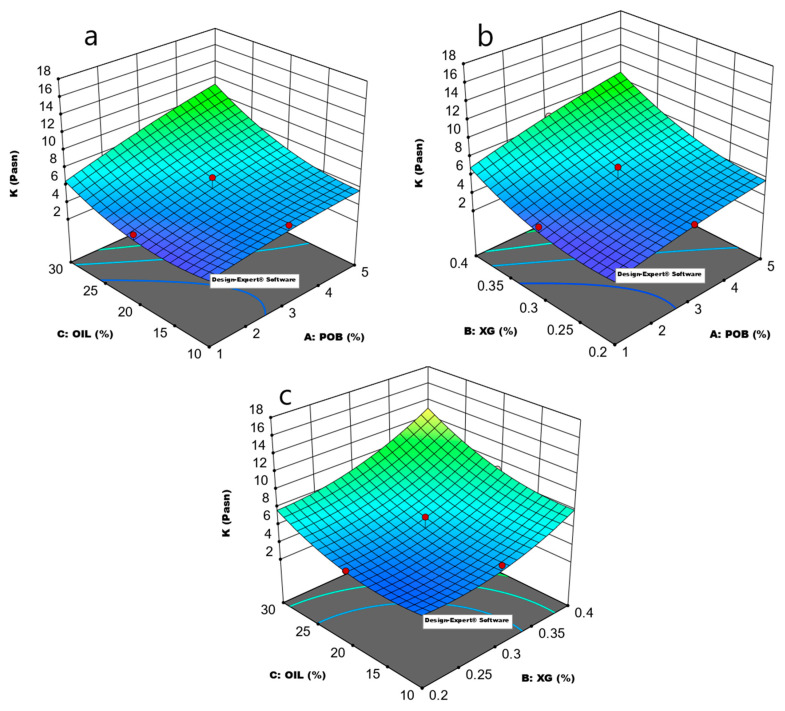
Response surface plot showing the effect of model parameters on the K value of salad dressing. (**a**): POB-Oil, (**b**): XG-POB, (**c**): XG-Oil (A: XG (xanthan gum), B: POB (pumpkin seed oil by-product), C: fat (milk fat), K: consistency coefficient).

**Figure 5 foods-10-02759-f005:**
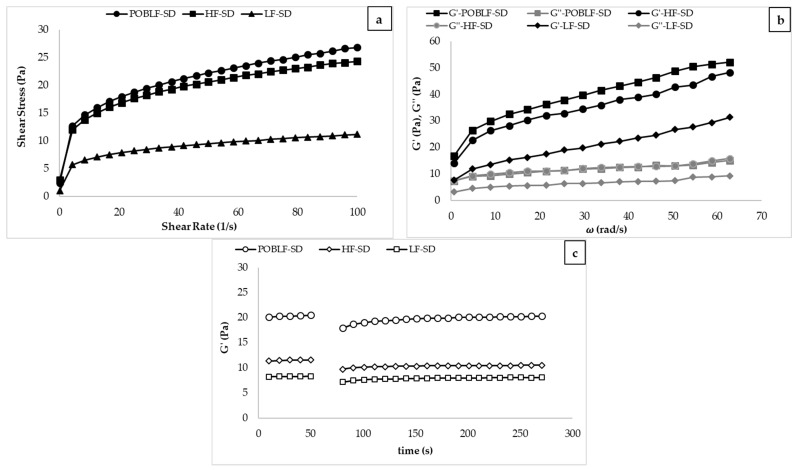
Rheological properties of POBLF-SD, HF-SD, LF-SD (HF-SD: high-fat salad dressing sample; LF-SD: low-fat salad dressing sample; POBLF-SD: low-fat salad dressing sample with cold-pressed pumpkin seed oil by-product; G′ (Pa): storage modulus, G″ (Pa): loss modulus). (**a**): The flow behavior of POBLF-SD, HF-SD, LF-SD; (**b**): The dynamic rheological behavior of POBLF-SD, HF-SD, LF-SD; (**c**): The 3-ITT rheological behavior of POBLF-SD, HF-SD, LF-SD.

**Figure 6 foods-10-02759-f006:**
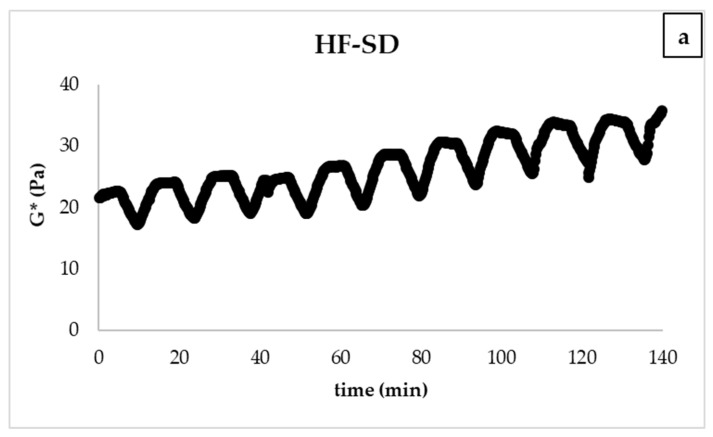
The change in G* value after applying the thermal loop (HF-SD: high-fat salad dressing sample; LF-SD: low-fat salad dressing sample; POBLF-SD: low-fat salad dressing sample with cold-pressed pumpkin seed oil by-product; G* (Pa): complex modulus). (**a**): thermal loop of HF-SD; (**b**): thermal loop of LF-SD; (**c**): thermal loop of POBLF-SD.

**Figure 7 foods-10-02759-f007:**
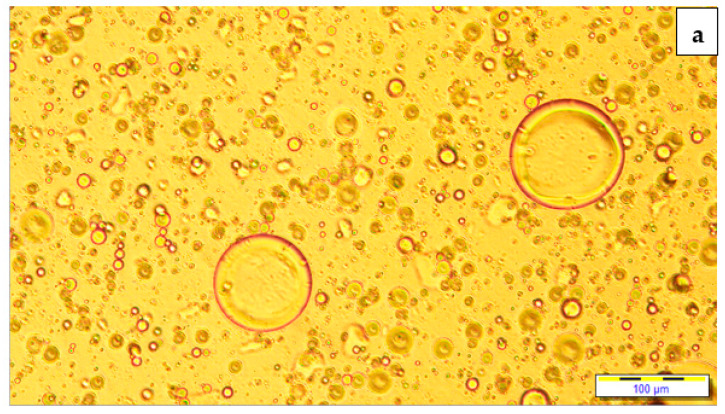
Light microscopy pictures of HF-SD (**a**), POBLF-SD (**b**), and LF-SD (**c**) (HF-SD: high-fat salad dressing sample; LF-SD: low-fat salad dressing sample; POBLF-SD: low-fat salad dressing sample with cold-pressed pumpkin seed oil by-product).

**Table 1 foods-10-02759-t001:** Steady shear power-law parameters of the salad dressing samples contained pumpkin seed by-products with ingredient compositions.

	POB (%)	XG (%)	EYP (%)	Oil (%)	K (Pa·s^n^)	n	R^2^
POB-1	3.0	0.3	3.00	30.0	8.60 ± 0.13	0.26 ± 0.00	1.00 ± 0.00
POB-2	1.0	0.4	3.00	30.0	9.86 ± 0.04	0.22 ± 0.00	1.00 ± 0.00
POB-3	3.0	0.4	3.00	20.0	9.09 ± 0.08	0.20 ± 0.00	0.99 ± 0.00
POB-4	1.0	0.2	3.00	30.0	5.60 ± 0.06	0.29 ± 0.00	1.00 ± 0.00
POB-5	1.0	0.4	3.00	10.0	6.41 ± 0.04	0.18 ± 0.00	1.00 ± 0.00
POB-6	1.0	0.3	3.00	20.0	4.60 ± 0.04	0.23 ± 0.00	0.99 ± 0.00
POB-7	3.0	0.2	3.00	20.0	5.05 ± 0.06	0.30 ± 0.00	1.00 ± 0.00
POB-8	3.0	0.3	3.00	10.0	7.84 ± 0.02	0.24 ± 0.00	0.99 ± 0.00
POB-9	3.0	0.3	3.00	20.0	5.12 ± 0.02	0.25 ± 0.00	1.00 ± 0.00
POB-10	5.0	0.4	3.00	30.0	16.11 ± 0.18	0.23 ± 0.00	1.00 ± 0.00
POB-11	5.0	0.2	3.00	30.0	9.56 ± 0.31	0.28 ± 0.01	1.00 ± 0.00
POB-12	3.0	0.3	3.00	20.0	5.16 ± 0.04	0.24 ± 0.00	1.00 ± 0.00
POB-13	1.0	0.2	3.00	10.0	3.75 ± 0.01	0.29 ± 0.00	1.00 ± 0.00
POB-14	3.0	0.3	3.00	20.0	5.14 ± 0.01	0.25 ± 0.00	1.00 ± 0.00
POB-15	5.0	0.4	3.00	10.0	8.95 ± 0.04	0.20 ± 0.00	0.99 ± 0.00
POB-16	5.0	0.2	3.00	10.0	5.95 ± 0.01	0.29 ± 0.01	1.00 ± 0.00
POB-17	5.0	0.3	3.00	20.0	6.54 ± 0.72	0.24 ± 0.01	1.00 ± 0.00

POB: cold-pressed pumpkin seed oil by-product; XG: xanthan gum; EYP: egg yolk powder; K (Pa·s^n^): consistency index values; n: the flow behavior index values.

**Table 2 foods-10-02759-t002:** Power-law parameters defining dynamic rheological properties of the salad dressing samples.

	POB (%)	XG (%)	Oil (%)	$G^{\prime }=K^{\prime }{\left(\omega \right)}^{n^{\prime }}$			$G^{\prime \prime }=K^{\prime \prime }{\left(\omega \right)}^{n^{\prime \prime }}$		
				K′	n′	R^2^	K″	n″	R^2^
POB-1	3.0	0.3	30.0	19.61 ± 0.31	0.28 ± 0.02	0.99 ± 0.00	7.98 ± 0.10	0.19 ± 0.00	0.98 ± 0.00
POB-2	1.0	0.4	30.0	24.07 ± 0.82	0.25 ± 0.01	0.99 ± 0.01	8.38 ± 0.27	0.20 ± 0.04	0.96 ± 0.02
POB-3	3.0	0.4	20.0	24.89 ± 0.78	0.25 ± 0.02	0.99 ± 0.00	9.28 ± 0.20	0.16 ± 0.00	0.98 ± 0.00
POB-4	1.0	0.2	30.0	4.32 ± 0.06	0.46 ± 0.01	0.98 ± 0.01	3.09 ± 0.02	0.23 ± 0.01	0.99 ± 0.00
POB-5	1.0	0.4	10.0	6.83 ± 0.04	0.37 ± 0.02	0.97 ± 0.01	3.11 ± 0.29	0.22 ± 0.00	0.93 ± 0.03
POB-6	1.0	0.3	20.0	5.06 ± 0.28	0.42 ± 0.03	0.98 ± 0.00	2.79 ± 0.46	0.27 ± 0.01	0.95 ± 0.02
POB-7	3.0	0.2	20.0	7.66 ± 0.08	0.36 ± 0.01	0.98 ± 0.01	3.94 ± 0.07	0.22 ± 0.02	0.99 ± 0.00
POB-8	3.0	0.3	10.0	8.69 ± 0.10	0.34 ± 0.06	0.98 ± 0.01	4.00 ± 0.02	0.20 ± 0.01	0.99 ± 0.00
POB-9	3.0	0.3	20.0	20.87 ± 0.08	0.20 ± 0.01	1.00 ± 0.00	7.31 ± 0.12	0.16 ± 0.01	0.95 ± 0.02
POB-10	5.0	0.4	30.0	93.40 ± 2.54	0.17 ± 0.03	1.00 ± 0.00	28.88 ± 1.16	0.12 ± 0.00	0.98 ± 0.01
POB-11	5.0	0.2	30.0	35.40 ± 0.41	0.22 ± 0.04	0.99 ± 0.00	11.97 ± 0.19	0.18 ± 0.02	0.96 ± 0.02
POB-12	3.0	0.3	20.0	22.06 ± 0.24	0.24 ± 0.02	0.99 ± 0.00	7.96 ± 0.09	0.17 ± 0.15	0.93 ± 0.03
POB-13	1.0	0.2	10.0	0.64 ± 0.09	0.81 ± 0.07	0.98 ± 0.01	0.94 ± 0.04	0.35 ± 0.01	0.94 ± 0.05
POB-14	3.0	0.3	20.0	15.80 ± 0.54	0.28 ± 0.01	0.99 ± 0.00	6.74 ± 0.52	0.17 ± 0.03	0.96 ± 0.02
POB-15	5.0	0.4	10.0	43.45 ± 1.07	0.18 ± 0.00	0.99 ± 0.00	17.23 ± 0.89	0.09 ± 0.00	0.99 ± 0.00
POB-16	5.0	0.2	10.0	13.18 ± 0.02	0.29 ± 0.01	0.98 ± 0.01	5.48 ± 0.18	0.20 ± 0.02	0.98 ± 0.00
POB-17	5.0	0.3	20.0	37.81 ± 0.35	0.21 ± 0.03	0.99 ± 0.00	13.12 ± 0.05	0.15 ± 0.02	0.95 ± 0.02

POB: cold-pressed pumpkin seed oil by-product; XG: xanthan gum; EYP: egg yolk powder (3%); G′(Pa): the storage modulus; G″ (Pa): the loss modulus; K′ and K″: consistency index values; n′ and n″: the flow behavior index values; ω(1/s): the angular velocity value.

**Table 3 foods-10-02759-t003:** Second-order structural kinetic model parameters for 3-ITT.

	POB (%)	XG (%)	Oil (%)	k	G_e_	G_0_	k × 1000	G_e_/G_o_	R^2^
POB-1	3.0	0.3	30.0	0.07 ± 0.01	29.76 ± 0.38	24.22 ± 0.54	67.80	1.23	0.98 ± 0.00
POB-2	1.0	0.4	30.0	0.05 ± 0.03	50.61 ± 2.69	42.23 ± 0.47	53.97	1.20	0.96 ± 0.02
POB-3	3.0	0.4	20.0	0.05 ± 0.00	29.64 ± 0.44	24.88 ± 0.00	52.27	1.19	1.00 ± 0.00
POB-4	1.0	0.2	30.0	0.04 ± 0.00	11.21 ± 0.23	9.67 ± 0.26	41.97	1.16	0.99 ± 0.00
POB-5	1.0	0.4	10.0	0.04 ± 0.01	14.20 ± 0.86	12.71 ± 0.30	36.54	1.12	1.00 ± 0.00
POB-6	1.0	0.3	20.0	0.04 ± 0.02	13.07 ± 0.54	11.67 ± 0.26	37.79	1.12	0.99 ± 0.00
POB-7	3.0	0.2	20.0	0.03 ± 0.00	11.20 ± 1.17	10.47 ± 0.65	27.44	1.07	0.99 ± 0.00
POB-8	3.0	0.3	10.0	0.03 ± 0.03	12.10 ± 0.10	10.90 ± 0.14	34.36	1.11	0.95 ± 0.04
POB-9	3.0	0.3	20.0	0.03 ± 0.00	26.45 ± 0.00	24.11 ± 0.00	33.55	1.10	0.99 ± 0.00
POB-10	5.0	0.4	30.0	0.13 ± 0.01	90.85 ± 1.03	60.19 ± 1.02	128.55	1.51	0.97 ± 0.00
POB-11	5.0	0.2	30.0	0.09 ± 0.02	82.09 ± 2.71	57.24 ± 2.09	87.49	1.43	0.97 ± 0.00
POB-12	3.0	0.3	20.0	0.03 ± 0.00	26.52 ± 0.00	24.40 ± 0.00	30.11	1.09	0.99 ± 0.00
POB-13	1.0	0.2	10.0	0.02 ± 0.00	5.02 ± 0.11	5.61 ± 0.22	23.30	0.89	0.97 ± 0.00
POB-14	3.0	0.3	20.0	0.03 ± 0.00	26.54 ± 0.00	24.32 ± 0.00	31.24	1.09	0.99 ± 0.00
POB-15	5.0	0.4	10.0	0.05 ± 0.00	27.18 ± 1.50	22.61 ± 0.51	54.78	1.20	0.98 ± 0.00
POB-16	5.0	0.2	10.0	0.03 ± 0.00	12.62 ± 0.42	11.64 ± 0.76	29.33	1.08	0.99 ± 0.00
POB-17	5.0	0.3	20.0	0.06 ± 0.04	32.87 ± 0.68	27.21 ± 0.00	57.63	1.21	0.98 ± 0.00

POB: cold-pressed pumpkin seed oil by-product; XG: xanthan gum; EYP: egg yolk powder (3%); k: the thixotropic rate constant; G_0_ (Pa): the initial storage modulus in the third time interval; G_e_: the equilibrium storage modulus.

**Table 4 foods-10-02759-t004:** The significance of the regression models established (F and *p* value) as a result of RSM.

Source	df	K		
		Mean Square	F-Value	*p*-Value (Prob > F)
Model	9	15.36	23.31	0.0002
Linear				
A-POB	1	25.25	38.31	0.0004
B-XG	1	46.27	70.20	< 0.0001
C-Oil	1	39.32	59.67	0.0001
Cross Product				
AB	1	1.65	2.50	0.1579
AC	1	5.23	7.94	0.0259
BC	1	2.15	3.27	0.1137
Quadratic				
A2	1	0.2030	0.3081	0.5961
B2	1	4.02	6.10	0.0429
C2	1	5.06	7.68	0.0276
Lack of Fit	5	0.4052	0.3132	0.8722
R-Squared				0.9677
Adj R-Squared				0.9262
Predicted R²				0.8450
Adeq Precision				18.7793

POB: cold-pressed pumpkin seed oil by-product; XG: xanthan gum; K, K′ and K″ (Pa·s^n^): consistency index values; df: degree of freedom. Values of “Prob > F” less than 0.1000 indicate model terms are significant.

**Table 5 foods-10-02759-t005:** The rheological properties, zeta potential, and particle size of HF-SD, LF-SD, and POBLF-SD.

Rheological Analysis		Samples		
		HF-SD	LF-SD	POBLF-SD
Steady shear	K (Pas^n^)	8.02 ^a^	3.78 ^b^	8.21 ^a^
σ = K × γ^n^	n	0.21 ^b^	0.23 ^a^	0.19 ^c^
	R^2^	0.99	0.99	0.99
Frequency				
	K′	13.80 ^b^	5.35 ^c^	15.78 ^a^
G′ = K′ × (ω)^n′^	n′	0.17 ^b^	0.36 ^a^	0.14 ^c^
	R^2^	1.00	0.99	0.98

G″ = K″ × (ω)^n″^	K″	5.74 ^a^	1.20 ^b^	6.16 ^a^
	n″	0.24 ^b^	0.37 ^a^	0.19 ^c^
	R^2^	0.99	0.92	0.93
3-ITT	G_0_	16.74 ^a^	6.87 ^b^	17.91 ^a^
	G_e_	20.65 ^b^	8.00 ^c^	22.85 ^a^
	k	0.05 ^b^	0.04 ^c^	0.06 ^a^
	G_e_/G_0_	1.23 ^b^	1.16 ^c^	1.28 ^a^
	k × 1000	45.01 ^b^	43.32 ^c^	56.64 ^a^
	R^2^	0.98	0.98	0.99
ζ-potential (mV)		−43.15 ± 0.93 ^b^	−39.68 ± 0.75 ^a^	−42.32 ± 0.68 ^b^
d_32_ (µm)		4904.83 ± 143.01 ^b^	5196.00 ± 65.87 ^a^	3125.67 ± 32.79 ^c^
PdI		0.90 ± 0.10 ^a^	0.27 ± 0.07 ^c^	0.65 ± 0.08 ^b^

HF-SD: high-fat salad dressing sample; LF-SD: low-fat salad dressing sample; POBLF-SD: low-fat salad dressing sample with cold-pressed pumpkin seed oil by-product (10% oil, 0.365% XG, 3.004% POB). Different lowercase letters in the same column indicate statistical differences between samples subjected to a different temperature (*p* < 0.05).

**Table 6 foods-10-02759-t006:** The oxidation kinetic parameters of the salad dressing samples.

Sample	Temperature (°C)	IP (h)	Ea (kJ/mol)	ΔH^++^ (kJ/mol)	ΔS^++^ (J/mol/K)	ΔG^++^ (kJ/mol)
HF-SD	80	12.57 ^aB^	86.63 ^B^	89.38 ^B^	6.61 ^B^	87.05
	90	3.20 ^bB^				86.98
	100	1.55 ^cB^				86.92
	110	1:03 ^dB^				86.85
LF-SD	80	10.34 ^aC^	76.69 ^C^	94.53 ^A^	22.35 ^A^	86.64
	90	2.58 ^bC^				86.41
	100	1.27 ^cC^				86.19
	110	0.45 ^dC^				85.97
POBLF-SD	80	16.25 ^aA^	88.83 ^A^	77.86 ^C^	−28.84 ^C^	88.04
	90	6.20 ^bA^				88.33
	100	2.46 ^cA^				88.62
	110	1.29 ^sA^				88.91

HF-SD: high-fat salad dressing sample; LF-SD: low-fat salad dressing sample; POBLF-SD: low-fat salad dressing sample with cold-pressed pumpkin seed oil by-product (10% oil, 0.365% XG, 3.004% POB). Different lowercase letters in the same column indicate statistical differences between samples subjected to a different temperature (*p* < 0.05). Different uppercase letters in the same column indicate statistical differences between samples HF-SD, LF-SD, and POBLF-SD (*p* < 0.05).

## Data Availability

The data presented in this study are available on request from the corresponding author.

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
