# Peer review of "The Potential Use of Cold-Pressed Pumpkin Seed Oil By-Products in a Low-Fat Salad Dressing: The Effect on Rheological, Microstructural, Recoverable Properties, and Emulsion and Oxidative Stability"

_foods, 2021, doi:10.3390/foods10112759_

Round 1
Reviewer 1 Report
- Please be sure that your manuscript thoroughly establishes how this work is fundamentally novel. This should include specific comparisons to previously published systems that have a similar purpose. Please present a strong case for how this work is a major advance.
- Line 102-103 Did the POB was ground before use? I guess the author should grind the POB via a grinder, please give the detailed information. Moreover, in order to maintain the uniform particle size of POB, did the POB was sieved via a mesh? If yes, please give the No. of mesh. As we known, the POB might contain some insoluble substance, the particle size was crucial for its solubility and dispersibility in the water.
- Except for the content of protein, oil and ash, how about the content of the other component? For example: fiber, carbohydrates, etc.
- Line 114, what the meaning of “and water was dispersed to water”?
- Line 115, the author said that “EYP and salt were dissolved in the dispersion”, what this dispersion meaning? Water??
- Can the author explain the reason for the determination of “Emulsion stability by thermal loop test”
- Line 208-209, why the temperature was adjusted to 80, 90, 100, and 110 °C?
Author Response
Open Review
Reviewer -1
Comments and Suggestions for Authors
- Please be sure that your manuscript thoroughly establishes how this work is fundamentally novel. This should include specific comparisons to previously published systems that have a similar purpose. Please present a strong case for how this work is a major advance.
Response: The novelty of this study was written in introduction part. Some studies have been conducted on the use of cold-pressed oil by-products in low-fat emulsions. However, in this study, formulation optimization was performed for the first time in low-fat salad dressings prepared with the cold pressed pumpkin seed oil by-product (POB). In addition, in this study, the effect of POB on oxidation stability of low-fat salad dressing was firstly investigated. In this study, thermal loop test, zeta potential and particle size distribution of low-fat salad dressing prepared with POB were also analyzed. In this study, a comprehensive characterization of low-fat salad dressing prepared with POB was performed.
- Line 102-103 Did the POB was ground before use? I guess the author should grind the POB via a grinder, please give the detailed information. Moreover, in order to maintain the uniform particle size of POB, did the POB was sieved via a mesh? If yes, please give the No. of mesh. As we known, the POB might contain some insoluble substance, the particle size was crucial for its solubility and dispersibility in the water.
Response: Mesh number and grinding machine were provided in material section.
- Except for the content of protein, oil and ash, how about the content of the other component? For example: fiber, carbohydrates, etc.
Response: The total carbohydrate content of POB (37.81 %) was provided.
- Line 114, what the meaning of “and water was dispersed to water”?
Response: The part related to preparation of salad dressing was rewritten.
- Line 115, the author said that “EYP and salt were dissolved in the dispersion”, what this dispersion meaning? Water??
Response: Response: The part related to preparation of salad dressing was rewritten.
Can the author explain the reason for the determination of “Emulsion stability by thermal loop test”
Response:
In this study, emulsion stability was also determined by the classical phase separation monitoring method. However, no phase separation was observed for 28 days for 3 samples. During thermal loop testing, environmental conditions are simulated with different temperature loop applications. In this way, the change in the dynamic rheological properties of the product can be detected numerically. we could not see phase separation using the classical phase separation method in the low-fat sample. However, thanks to the thermal loop test, a sudden increase in the G* value of the product was observed after 10 loops. This indicates that the stability of the low-fat sample will be lower in long-term storage. In the sample we prepared with POB, there was no sudden change in the G* value during 10 loops. Thermal loop test showed us that POBLF-SD sample is more stable than LF-SD sample. In short, the thermal loop test can quickly provide an idea about the stability of the emulsion in a longer period.
- Line 208-209, why the temperature was adjusted to 80, 90, 100, and 110 °C?
Response: The accelerated oxidation test was carried out with the OXI-TEST system. The reference method for OXI-TEST is 90°C. Using the IP value at this temperature, the oxidation stability between samples could be compared. In addition to using 90°C, in this study, OXITEST analysis was carried out at four different temperature in order to make oxidation kinetics. The maximum adjustable temperature of the device is 110 °C. We used 4 different temperatures, including the maximum operating temperature, to study more comprehensive oxidation stability by obtaining Arrhenius and activated complex parameters. Oxidation kinetics were performed at 4 different temperature points. We think that these 4 different temperatures will be sufficient to determine the degradation kinetics. If Reviewer recommends that we choose at 5 different points, we can work at 70 C in addition to these temperatures.

Reviewer 2 Report
Manuscript ID: foods-1440198
Type of manuscript: Article
Title: The Potential Use of Cold-Pressed Pumpkin Seed By-products as a Natural Fat Substitute and Stabilizer in a Low-Fat Salad Dressing
Authors: Zeynep Hazal Tekin-Cakmak, Ilker Atik, and Salih Karasu
Review of the manuscript
Comments
The manuscript presented for revision is very interesting. This work concerns an important area of science and importand for food processing at the same time - waste and by-products management. But methodology of experiment requires supplementation.
The most important comments to the presented manuscript:
- One of the most important attributes of food products, including salad dressing, are sensory characteristics. It is a pity, that there is nothing here about the taste, color or mouthfeel of the tested product variants. In the title of the article, it should be noted that the rheological, microstructural, recoverable properties, and emulsion and oxidative stability (or analysis of selected attributes) are investigated.
- l.46 „Food products that contained less fat may be undesirable for consumers because fat contributes to the texture, appearance, flavor, and the enhancing shelf life of emulsions [5, 6].” Fat affects shelf life of emulsions, but does it always enhance this shelf life? The emulsion in which fat rich in PUFA (e.g. linseed oil), has been used will behave completely differently, than emulsion with very oxidative stability oil (e.g. olive oil, sesame oil).
- l.74-80 The information, presented by the authors in this fragment of the manuscript relates to the composition of cold-pressed pumpkin oil. But in the expeller (cold-pressed pumpkin cake - probably this product was used in the experiment), which was added in the amount of 1-5%, only 9% of fat remained. How did this amount of fat change the nutritional value of fat in the finished salad dressing?
- l.101 2.1. Material
Could you provide more information on by-product (POB), used in this experiment. Are these expellers / cake after cold pressing pumpkin oil? Under what conditions was this expellers made, were the pumpkin seeds pressed in the shell, or shellless varieties ( Cucurbita pepo var. oleifera) ?
- l.109 2.2.1. Salad Dressing Preparation
Information on the preparation of salad dressing is unclear. Why was such a model and such a procedure used for making salad dressing?
- l. 113 – „POB was added to the dispersion after XG, and water was dispersed to water” - this is unclear
- l.116 – „The mixture was stirred at 1,000 rpm by a magnetic stirrer for 6 h until the hydration of XG was obtained” - why was hydrated of xantan gum so long? Why was such a high dose (0,2-0,4%) of XG used?
- The pH has a great influence on the formation of the emulsion and its stability. When were acidifying agents added (what was used?) and what was the pH of the emulsion. It,s very important!
- l.204. Oxidative stability by OXITEST
Why is oxidative stability tested only at four different temperatures (80, 90, 100 and 110 C)? The result would be more accurate and reliable, if the oxidative stability was also tested at anather temperature e.g. 85 or 95 C [SYMONIUK E., Ratusz K. and Krygier K., “Comparison of the oxidative stability of linseed (Linum usitatissimum L.) oil by pressure differential scanning calorimetry and Rancimat measurements.”, J. Food Sci. Technol., 53, 3986-3995, 2016; AKSOY, FS, et al., Oxidative stability of the salad) dressing enriched by microencapsulated phenolic extracts from cold-pressed grape and pomegranate seed oil by-products evaluated using OXITEST. Food Science and Technology, 2021..,]
- 3.1.5. Analysis of HF-SD, LF-SD, and POBLF-SD
What was the composition of the reference sample (HF-SD_High-fat salad dressing) sample (30% of oil and 0.35% XG), and pH?.
Author Response
Open Review
Reviewer -2
Comments
The manuscript presented for revision is very interesting. This work concerns an important area of science and important for food processing at the same time - waste and by-products management. But methodology of experiment requires supplementation.
The most important comments to the presented manuscript:
- One of the most important attributes of food products, including salad dressing, are sensory characteristics. It is a pity, that there is nothing here about the taste, color or mouthfeel of the tested product variants. In the title of the article, it should be noted that the rheological, microstructural, recoverable properties, and emulsion and oxidative stability (or analysis of selected attributes) are investigated.
Response: The title was revised according to reviewer suggestions. We did not want to keep the number of data too high, as there are many analyzes such as rheological analyzes, microstructural analyzes, formulation optimization and oxidative stability analysis, oxidation kinetics in the article. The revision time was also very limited. That's why we revised the title. However, if the referee's suggestion continues, we try to add textural analysis or sensory analysis within the revision period.
- l. 46„Food products that contained less fat may be undesirable for consumers because fat contributes to the texture, appearance, flavor, and the enhancing [5, 6].” Fat affects shelf life of emulsions, but does it always enhance this shelf life? The emulsion in which fat rich in PUFA (e.g. linseed oil), has been used will behave completely differently, than emulsion with very oxidative stability oil (e.g. olive oil, sesame oil).
Response: The shelf life was revised as emulsion stability. Emulsion stability is often improved with an increase in the oil phase volume fraction due to the increase in the packing fraction of oil droplets, as the droplets are more closely packed and the creaming/sedimentation rates are therefore lowered. When the oil content is reduced below a certain critical level in reduced-fat salad dressing or mayonnaise, the emulsion tends to become highly unstable. The reduction of fat in salad dressing and mayonnaise could have an impact on their packing characteristics which therefore could change their rheological and textural behavior and has also profoundly influence their stability during storage.
- l.74-80 The information, presented by the authors in this fragment of the manuscript relates to the composition of cold-pressed pumpkin oil. But in the expeller (cold-pressed pumpkin cake - probably this product was used in the experiment), which was added in the amount of 1-5%, only 9% of fat remained. How did this amount of fat change the nutritional value of fat in the finished salad dressing?
Response: 3.04% POB was used in the preparation of the POBLF-SD sample. In the total product formulation, the oil content from POB is approximately 0.27%. We anticipate that this amount will not make a significant change on the nutritional value of the product. Heat treatments during removal of the chemical solvent will cause dramatic changes in the nutritional value of the cold press by-product. This waste will no longer be cold press but solvent extraction waste. Therefore, its oil has not been removed with chemical solvents to preserve nutritional value.
- Could you provide more information on by-product (POB), used in this experiment. Are these expellers/cake after cold pressing pumpkin oil? Under what conditions was this expellers made, were the pumpkin seeds pressed in the shell, or shellless varieties (Cucurbita pepo var. oleifera) ?
Response: The information pumpkinseed and extraction conditions were provided follow: POB is the press cake released after cold press extraction of pumpkin seed (Cucurbita pepo L). The shells of the pumpkin seeds were removed before pressing. The press temperature did not exceed 50 °C during the cold-press process.
- Information on the preparation of salad dressing is unclear. Why was such a model and such a procedure used for making salad dressing?
Response: This procedures was rewritten and method was provided.
- 113 – „POB was added to the dispersion after XG, and water was dispersed to water”- this is unclear
Response: Salad dressing preparation procedure section was rewritten and misunderstandings in this section were tried to be resolved.
- l.116 – „The mixture was stirred at 1,000 rpm by a magnetic stirrer for 6 h until the hydration of XG was obtained” - why was hydrated of xantan gum so long? Why was such a high dose (0,2-0,4%) of XG used?
Response: This part was rewritten. It was reported in the literature that it cannot provide sufficient consistency below 0.35 percent of XG (Zhen Ma & Joyce I. Boye). In our study, products containing lower XG (0.2%) did not show sufficient viscoelastic solid character and pseudoplastic behaivor . That's why XG was used around 0.35%.
Zhen Ma & Joyce I. Boye. Advances in the Design and Production of Reduced-Fat and Reduced-Cholesterol Salad Dressing and Mayonnaise: A Review. Food and Bioprocess Technology volume 6, pages648–670 (2013).
- The pH has a great influence on the formation of the emulsion and its stability. When were acidifying agents added (what was used?) and what was the pH of the emulsion. It,s very important!
Response: The preparation of salad dressing part was rewritten. 7.5% of the aqueous phase consists of vinegar. The pH of the salad dressing samples was lower than 4.
- 204. Oxidative stability by OXITEST: Why is oxidative stability tested only at four different temperatures (80, 90, 100 and 110 C)? The result would be more accurate and reliable, if the oxidative stability was also tested at anather temperature e.g. 85 or 95 C [SYMONIUK E., Ratusz K. and Krygier K., “Comparison of the oxidative stability of linseed (Linum usitatissimum L.) oil by pressure differential scanning calorimetry and Rancimat measurements.”, J. Food Sci. Technol., 53, 3986-3995, 2016; AKSOY, FS, et al., Oxidative stability of the salad) dressing enriched by microencapsulated phenolic extracts from cold-pressed grape and pomegranate seed oil by-products evaluated using OXITEST. Food Science and Technology, 2021..,].
Response: The accelerated oxidation test was carried out with the OXI-TEST system. The reference method for OXI-TEST is 90°C. Using the IP value at this temperature, the oxidation stability between samples could be compared. In addition to using 90°C, in this study, OXITEST analysis was carried out at four different temperature in order to make oxidation kinetics. The maximum adjustable temperature of the device is 110 °C. We used 4 different temperatures, including the maximum operating temperature, to study more comprehensive oxidation stability by obtaining Arrhenius and activated complex parameters. Oxidation kinetics were performed at 4 different temperature points. We think that these 4 different temperatures will be sufficient to determine the degradation kinetics. If Reviewer recommends that we choose at 5 different points, we can work at 70 C in addition to these temperatures.
- 1.5. Analysis of HF-SD, LF-SD, and POBLF-SD: What was the composition of the reference sample (HF-SD_High-fat salad dressing) sample (30% of oil and 0.35% XG), and pH?.
Response: The composition of reference samples were provided.

Round 2
Reviewer 1 Report
The author have carefully revised the paper according to my opinion, and I think it should be accepted by the journal.
Reviewer 2 Report
Thank you very much for the changes made. These explanations and additional information will help readers to understand your experiment.